

# Sensitivity of snow density and specific surface area measured by microtomography to different image processing algorithms

Pascal Hagenmuller[1,2], Margret Matzl[3], Guillaume Chambon[2], and Martin Schneebeli[3]

[1]Météo-France - CNRS, CNRM-GAME, UMR 3589, CEN, 38400 Saint Martin d'Hères, France
[2]Irstea, UR ETGR Erosion torrentielle, neige et avalanches, 38402 Saint Martin d'Hères, France
[3]WSL Institute for Snow and Avalanche Research SLF, Fluelastrasse 11, 7260 Davos Dorf, Switzerland

*Correspondence to:* Pascal Hagenmuller (pascal.hagenmuller@meteo.fr)

**Abstract.** Microtomography can measure the X-ray attenuation coefficient in a 3D volume of snow with a spatial resolution of a few microns. In order to extract quantitative characteristics of the microstructure, such as the specific surface area (SSA), from these data, the grayscale image first needs to be segmented into a binary image of ice and air. Different numerical algorithms can then

5 be used to compute the surface area of the binary image. In this paper, we report on the effect of commonly used segmentation and surface area computation techniques on the evaluation of density and specific surface area. The evaluation is based on a set of 38 X-ray tomographies of different snow samples without impregnation, scanned with an effective voxel size of 10 and 18 $\mu$m. We found that different surface area computation methods can induce relative variations up to 5% in the

10 density and SSA values. Regarding segmentation, similar results were obtained with sequential and energy-based approaches provided the associated parameters are correctly chosen. The voxel size also appears to affect the values of density and SSA, but because images with the higher resolution also show the higher noise level, it was not possible to draw a definitive conclusion on this effect of resolution. Finally, practical recommendations concerning the processing of X-ray tomographic

15 images of snow are provided.

## 1 Introduction

The specific surface area (SSA) of snow is defined as the area $S$ of the ice-air interface per unit mass $M$, i.e. $SSA = S/M$ expressed in m$^2$ kg$^{-1}$. This quantity is essential for the modelling of the physical and chemical properties of snow because it is an indicator of potential exchanges with the



surrounding environment. For instance, SSA can be used to predict snow electromagnetic characteristics such as light scattering and absorption (albedo in the near infrared) (e.g. Warren, 1982; Flanner and Zender, 2006) or microwave radiation (e.g. Brucker et al., 2011), and snow metamorphism (e.g. Flin et al., 2004; Domine et al., 2007). Precise knowledge of this quantity is required in numerous applications such as cold regions hydrology, predicting the role of snow in the regional/global climate system, optical and microwave remote sensing, snow chemistry, etc.

In the last decade, numerous field and laboratory instruments were developed by different research groups to measure snow grain size. One possible definition of this grain size is the equivalent spherical radius $r_{eq}$, computed from the SSA as $r_{eq} = 3/(\rho_{ice} \times SSA)$ with $\rho_{ice}$ the density of ice. This definition is actually equivalent to the optical radius, i.e. the radius of a collection of spheres with the same infrared albedo as that of the snow microstructure (Warren, 1982). In contrast, the definition of grain size used in traditional snow classification as the mean of the longest extension of disaggregated particles (Fierz et al., 2009) is correlated to SSA only for a few snow classes. The crystal size as stereologically measured by Riche et al. (2012) is equivalent to $r_{eq}$ only for monocrystalline snow grains. Because of the co-existence of these inconsistent grain size definitions and of different associated measurement methods (optical, gas adsorption, tomography, stereology), an intercomparison of different grain size measurement methods was organised by the International Association of Cryospheric Sciences (IACS) working group "From quantitative stratigraphy to microstructure-based modelling of snow". One of the main objectives of this intercomparison was to determine the accuracy, comparability and quality of existing measurement methods.

In the context of this workshop, the present study focuses on the measurement of density and SSA derived from microtomographic data (Cnudde and Boone, 2013). Microtomography measures the X-ray attenuation coefficient in a three-dimensional (3D) volume with high spatial resolutions of a few to a few tens of microns. To extract the microstructure from the reconstructed 3D image, this image has to be filtered to reduce noise and subsequently analysed to identify the phase relevant for the investigation, in our case ice. This step, called binary segmentation, affects the subsequent microstructure characterisation, especially when the image resolution is close to the typical size of microstructural details. Different algorithms then exist to calculate density and SSA from this images. Here we investigate the effects of binary segmentation and surface area calculation methods on density and SSA estimates, in order to provide guidelines for the use of snow sample microtomographic data.

Comparative studies of processing techniques for images obtained via X-ray microtomography have already listed the performance of several segmentation methods with respect to different quality indicators (e.g. Kaestner et al., 2008; Iassonov et al., 2009; Schlüter and Sheppard, 2014). These studies emphasised the importance of using local image information such as spatial correlation to perform suitable segmentations and highlighted the superior performance of Bayesian Markov random field segmentation (Berthod et al., 1996), which consists in finding the segmentation with min-



imum boundary surface and which at the same time respects the gray value data in the best possible way. However, none of the mentioned studies were interested in snow. This material exhibits specific features such as a natural tendency, induced by metamorphism, to minimise its surface energy (e.g.

Flin et al., 2003; Vetter et al., 2010). Moreover, these former studies tended to focus on properties linked to volumetric material contents, while less attention was paid to the surface area of the segmented object. Hagenmuller et al. (2013) applied an energy-based segmentation method on images of impregnated snow samples, which is a three-phase material (impregnation product, ice and residual air bubbles). This method is based on the same principles as the Bayesian Markov random field

segmentation but the optimisation process is performed differently. It takes explicitly advantage of the knowledge that the surface energy of snow tends to be minimal, and was shown to be accurate in comparison to a segmentation method based on global thresholding. However the set of snow microtomographic images used by Hagenmuller et al. (2013) was limited to impregnated samples and to a few different snow types. Moreover, no independent SSA measurements were available to

provide a reference or at least a comparison. Here, the flexible energy-based segmentation method was adapted to two-phase images (air-ice) and applied to 38 images on which SSA measurements were conducted with independent instruments. Note that comparisons with these independent SSA measurements are beyond the scope of this paper and will be reported in a synthesis paper of the working group to be published in the present special issue of The Cryosphere.

First, the sampling and X-ray measurement procedures to obtain grayscale images are described. Attention is paid to the fact that the parameters used for binary segmentation also depend on the scanned sample and not only on the X-ray source setup. Second, two different approaches of binary segmentation are presented. The first one, commonly used in the snow science community, consists of a sequence of filters: Gaussian smoothing, global thresholding and morphological filtering.

The second one is based on the minimisation of a segmentation energy. Third, different methods to compute surface area from binary images are presented. Finally, the different methods of binary segmentation and area computation are applied to the microtomographic images and the results are compared to provide an estimation of the scatter in density and SSA measurements due to numerical processing of the grayscale image and area calculation.

## 2 Material and methods

### 2.1 Data set

Snow sampling, preparation and scanning were conducted at SLF, Davos, Switzerland, during the Snow Grain Size Workshop in March 2014.





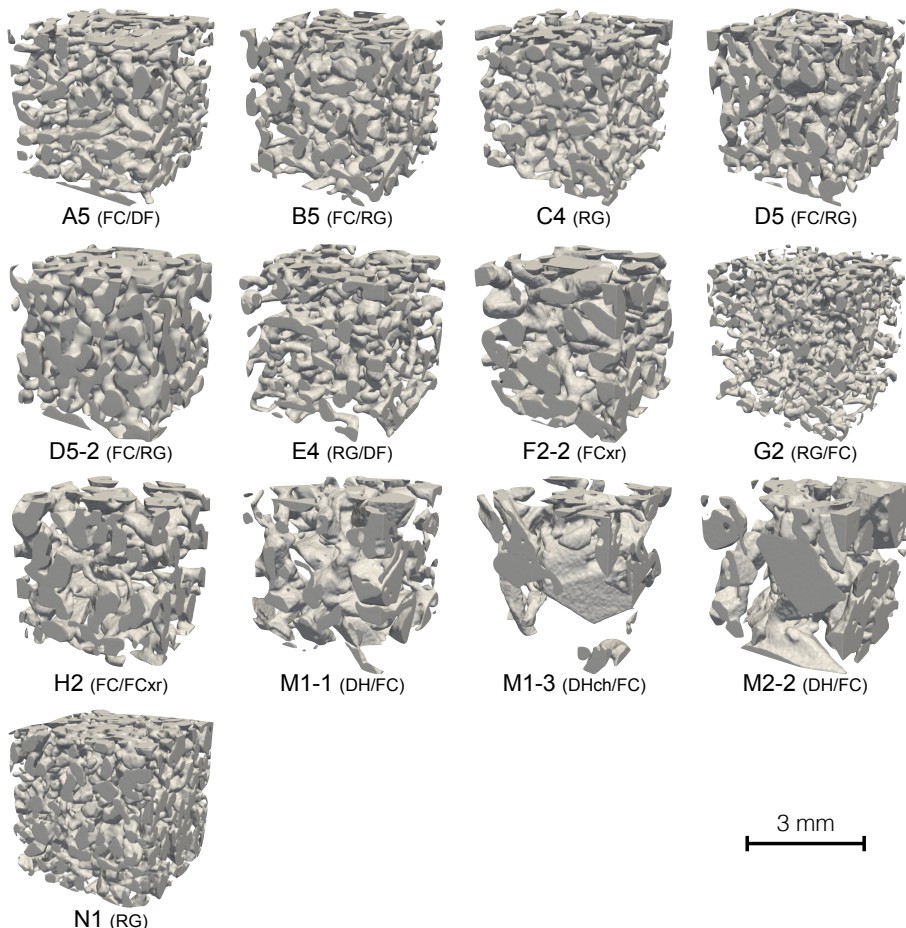

**Figure 1.** The different snow types and microstructural patterns used in this study. The 3D images shown have a side-length of 3 mm and correspond to a subset of the images analysed in the present study. The grain shape is also indicated in brackets below the images, according to the international snow classification (Fierz et al., 2009).

### 2.1.1 Sampling

Thirteen snow blocks of apparently homogeneous snow were collected in the field or prepared in a cold laboratory. These blocks span different snow types (decomposing and fragmented snow, rounded grains, faceted crystals and depth hoar, Fig. 1). Smaller specimens were taken out of these blocks to conduct grain size measurements with different instruments. Two snow cylinders of radius 35 mm and height of 60 mm and one snow cylinder of radius 20 mm and 60 mm height were

extruded from each block to perform microtomographic measurements.





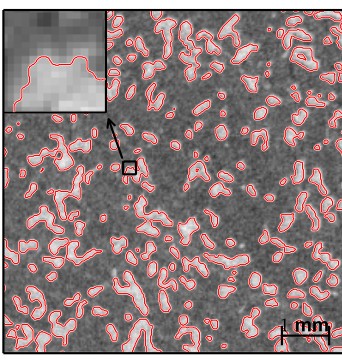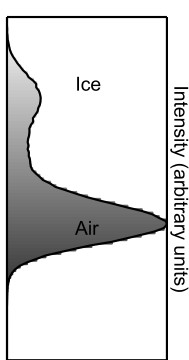

**Figure 2.** Grayscale image ($400^2$ pixels) representing the X-ray attenuation coefficient and its corresponding grayscale histogram. The 2D slice is extracted from image G2-s1. The image exhibits two materials: air (dark gray) and ice (light gray). The contour of ice resulting from binary segmentation is plotted in red. The zoom panel (top right) was enlarged eight times to emphasise the fuzzy transition between air and ice.

### 2.1.2 X-ray scanning

The grayscale images were obtained with a commercial microcomputer tomograph (Scanco Medical $\mu$CT40) operating in a cold room at $-15°C$. The X-ray source was set to an energy of 55 keV. The two samples with a radius of 37 mm were scanned with a nominal resolution of 18 $\mu$m, and the smaller sample with a radius of 20 mm was scanned with a nominal resolution of 10 $\mu$m. To avoid edge effects a sub image of size about $1000^3$ voxels was extracted from each image, which correspond to a volume of $10^3$ mm$^3$ for the 10 $\mu$m resolution and $18^3$ mm$^3$ for the 18 $\mu$m resolution. These volumes are larger than the previously established representative elementary volumes for SSA (around $2.5^3$ mm$^3$, Flin et al. (2011)) and density (around $2.5^3$ mm$^3$, Coléou et al. (2001)). In the following, the images corresponding to a resolution 18 $\mu$m are identified by the suffixes "s1" and "s2", and the 10 $\mu$m by "10micron". The output of the tomograph is a 3D grayscale image with values encoded as unsigned short integers (16 bits) (Fig. 2). The grayscale value quantifies the X-ray attenuation coefficient.

### 2.1.3 Images artefacts

As shown in Fig. 2, air and ice can be distinguished by their respective attenuation coefficient, i.e. by their grayscale value or intensity. However, the grayscale distributions in ice and air are not completely disjoint, as there are always pixels (voxels in 3d) that consist of both materials. In addition to this fuzzy transition between air and ice, the image is also noisy, which makes the binary segmentation not straightforward.





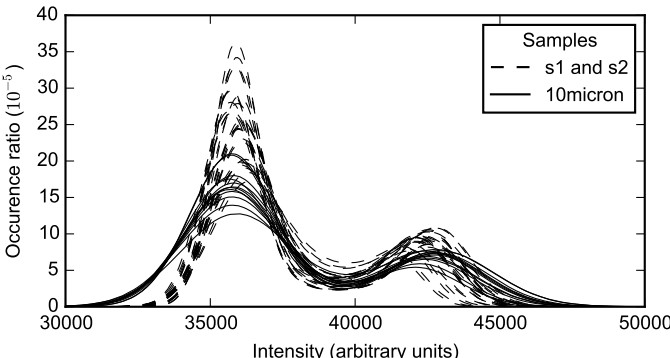

**Figure 3.** Grayscale distributions for all images. The solid lines and dashed lines respectively represent the distributions for the images scanned with the resolution 10 $\mu$m and 18 $\mu$m. The grayscale distribution is computed on 1000 bins of homogeneous size in the intensity range $[30000, 50000]$ (arbitrary units).

Figure 3 shows the grayscale distributions obtained on all scanned images. The exact position of the attenuation peaks and the scatter around the peaks depend both on resolution and snow sample. Slight differences are also observed between the grayscale distribution of the two images coming from the same snow block and scanned with the same resolution. This may be due slight variations in the temperature of the X-ray source during successive scans. Hence, it is doubtful whether binary segmentation parameters "optimised" for one image can be used to segment other images even of the same sample and with the same resolution. It appears necessary to determine the segmentation parameters on each image independently.

## 2.2 Segmentation methods

In this section, two binary segmentation methods are presented: (1) the common method based on global thresholding combined with denoising and morphological filtering, hereafter referred to as *sequential filtering*, and (2) a method based on the minimisation of a segmentation energy, referred to as *energy-based segmentation*.

### 2.2.1 Sequential filtering

Sequential filtering is commonly used to segment grayscale microtomographic images of snow because it is simple, fast, and is implemented in packages of several different programming languages. It consists of a sequence of denoising, global thresholding and post-processing, the input of each step coming from the output of the previous step.

*Denoising with a Gaussian filter.* Numerous filters exist to remove noise from images, the most common being the Gaussian filter, the median filter, the anisotropic diffusion filter and the total





variation filter (Schlüter and Sheppard, 2014). The objective of denoising is to smooth intensity variations in homogeneous zones (characterised by low-intensity gradients) while preserving sharp variations of intensity in the transition between materials (characterised by high-intensity gradient). In snow science, the most popular denoising filter is the Gaussian filter (e.g. Kerbrat et al., 2008; Lomonaco et al., 2011; Theile et al., 2009; Schleef and Löwe, 2013), which consists in convoluting the intensity field $I$ with a Gaussian kernel of zero mean $\mathcal{N}(0,\sigma)$ defined as:

$$\mathcal{N}(0,\sigma)(I) = \frac{1}{\sigma\sqrt{2\pi}}\exp\left(-\frac{I^2}{2\sigma^2}\right) \tag{1}$$

with $\sigma$ the (positive) standard deviation. The support of the Gaussian kernel can be truncated (here to $\lfloor 4\sigma \rfloor$) to speed up the calculations. This filter is very efficient in smoothing homogeneous zones. However it fails to preserve sharp features in the image by indifferently smoothing low-intensity and high-intensity gradient zones, and therefore reduces the effective resolution of the image.

*Global thresholding.* After the denoising step, a global threshold $T$ is determined for the entire image in order to classify voxels as air or ice depending whether their grayscale value is smaller or greater, respectively, than the threshold. The choice of this threshold is generally based on the grayscale histogram without considering the spatial distribution of grayscale values. Different methods exist, an exhaustive review of which can be found in Sezgin and Sankur (2004). Here the focus is limited to methods commonly used for snow, namely: (1) local minimum, (2) Otsu's method, and (3) mixture modelling.

1. *Local minimum:* A simple way to determine the threshold is to define it as the local minimum in the valley between the attenuation peaks of ice and air (Fig. 3) (e.g. Flin, 2004; Heggli et al., 2009; Pinzer et al., 2012). However, the histogram may be noisy, resulting in several local maxima and minima, which makes the method inapplicable. In some cases, the attenuation peaks of ice and air can also be too close, which results in a unimodal histogram without any valley (e.g. Kerbrat et al., 2008). Moreover, the position of the local minimum is generally affected by the height of the attenuation peaks in the histogram: the less ice in the image, the closer the local minimum is to the ice attenuation peak. The threshold obtained with this method is denoted $T_{valley}$ in the following.

2. *Otsu's method:* Another popular method, first introduced by Otsu (1975), is to find the threshold that minimises the intra-class variance $\sigma_w$ defined as $\sigma_w^2 = n_{air}\sigma_{air}^2 + n_{ice}\sigma_{ice}^2$, with $n_{air}$ and $n_{ice}$ the numbers of voxels classified as air and ice respectively, and $\sigma_{air}$ and $\sigma_{ice}$ the standard deviations of the grayscale value in each segmented class. This method is generic and does not require any assumption on the grayscale distribution. However, this also represents a drawback of the method, in that knowledge of the origin of the image artefacts can help to find the optimal threshold. The threshold obtained with this method is denoted $T_{otsu}$ in the following. This method is less used in the snow community but is widely used for other porous materials (e.g. Haussener, 2010; Ebner et al., 2015).





3. *Mixture modelling:* The classification error induced by the thresholding can be also minimised by assuming that each class is Gaussian-distributed. From there, different methods can be considered to decompose the grayscale histogram in a sum of Gaussian distributions:

   – The grayscale histogram computed on the image masked on high-intensity gradient can usually be perfectly decomposed into two Gaussian distributions centred on $\mu_{air}$ and $\mu_{ice}$, respectively, and with identical standard deviations $\sigma$ (Fig. 4a). Masking the high-intensity gradients enables to suppress the fuzzy transition zones between ice and air. Therefore, on the corresponding histogram, the scatter around the attenuation peaks can be attributed to instrument noise only, and appears to be Gaussian distributed. The optimal threshold value derived from this method is $T_{mask} = (\mu_{ice} + \mu_{air})/2$. Note that the ratio $Q_{noise} = \sigma/(\mu_{ice} - \mu_{air})$ provides a quantitative estimate of the quality of the grayscale images with regards to noise artefacts. In practice, however, masking the grayscale image on high-intensity gradient zones is time consuming and not straightforward with existing segmentation softwares. Moreover, in case of very thin ice structures, homogeneous ice zones are almost inexistent.

   – Kerbrat et al. (2008) directly fitted the sum of two Gaussian distributions to the complete grayscale histogram (Fig. 4b). Note that the partial volume effect at the transition between materials changes the position of the attenuation peaks and the agreement between the fit and the histogram remains partial in comparison to the fit on the histogram of the masked image. The threshold, defined as the mean of the centre of the two Gaussian distributions obtained with this fit, is denoted $T_{kerbrat}$.

   – Hagenmuller et al. (2013) fitted the grayscale histogram with the sum of three Gaussian distributions to take into account the fuzzy transition between materials. The fitting function is

$$\tilde{F} = \lambda_{air} \cdot \mathcal{N}(\mu_{air}, \sigma) + \lambda_{ice} \cdot \mathcal{N}(\mu_{ice}, \sigma) + (1 - \lambda_{air} - \lambda_{ice}) \cdot \mathcal{N}(\bar{\mu}, \tilde{\mu}) \qquad (2)$$

   where $\lambda_{air}$, $\lambda_{ice}$, $\mu_{air}$, $\mu_{ice}$ and $\sigma$ are five adjustable parameters, $\bar{\mu} = (\mu_{air} + \mu_{ice})/2$, and $\tilde{\mu} = (\mu_{ice} - \mu_{air})/4$. The two first terms of the sum model the grayscale distribution in low-intensity gradient zones, while the last term models the grayscale distribution in high-intensity gradient zones. The choice of $\tilde{\mu}$-value is arbitrary but was found to provide a good fit to the transition zone. The agreement of this model with the grayscale histogram is generally very good, although no additional free parameter is added in comparison to the two-Gaussian distributions model (Fig. 4c). The optimal threshold value derived from this method is $T_{hagen} = (\mu_{ice} + \mu_{air})/2$. Note that the value $Q_{blur} = 1 - \lambda_{air} - \lambda_{ice}$ provides a quantitative estimate of the quality of the grayscale image with regards to the fuzzy transition artefact.





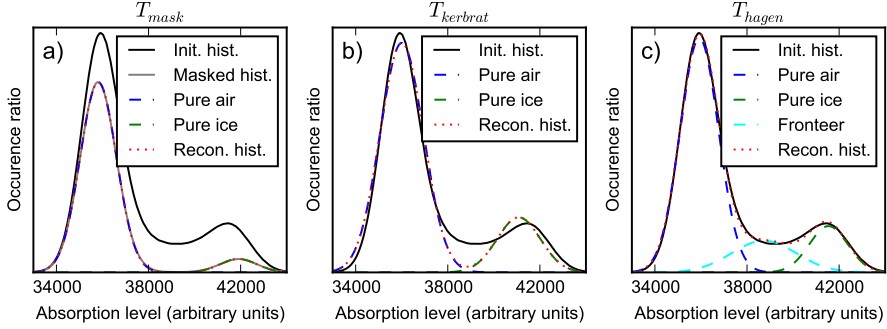

**Figure 4.** Different intensity models based on the grayscale distribution. (a) Mixture model composed of two Gaussian distributions to reproduce the grayscale distribution on the low-intensity gradient zones. $L_1$ error is 0.006. (b) Mixture model composed of two Gaussian distributions to reproduce the whole grayscale distribution. $L_1$ error is 0.14. (c) Mixture model composed of three Gaussian distributions to reproduce the whole grayscale distribution. $L_1$ error is 0.03. For all figures, the $L_1$ error is the integral of the absolute difference between the measured and the modelled grayscale distribution. Note that the area under the grayscale distribution of the entire image is 1.

*Post-processing.* In general, the binary segmented image needs to be further corrected to remove remaining artefacts. This can be done manually for each 2D section, but it is extremely time-consuming (Flin et al., 2003). The continuity of the ice matrix can also be used to correct the binary image by deleting ice zones not connected to the main structure or to the edges of the image (Hagenmuller et al., 2013; Schleef et al., 2014; Calonne et al., 2014). Among generic and automatic post-processing methods, the morphological operators erosion and dilation are the most popular. The combination of these operators enables to delete small holes in the ice matrix (closing: erosion then dilation) or small protuberances on the ice surface (opening: dilation then erosion). In the following, the support size if these morphological filters is denoted $d$.

### 2.2.2 Energy-based segmentation

Energy-based segmentation methods consist in finding the optimal segmentation by minimising a prescribed energy function. These methods are robust and flexible since the best segmentation is automatically found by the optimisation process, and the energy function can incorporate various segmentation criteria. In general, the optimisation of functions composed of billions of variables can be complex and time-consuming. However, provided that the variables are binary and some additional restrictions on the form of the energy function, efficient global optimisation methods exist. In particular, functions that involve only pair interactions can be globally optimised in a very efficient way with the graph cut method (Kolmogorov and Zabih, 2004). Using this method, the





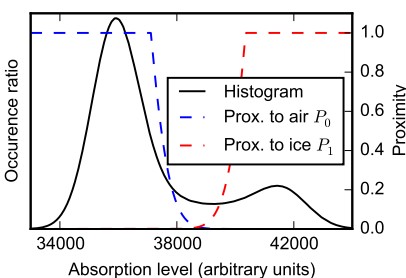

**Figure 5.** Proximity functions to air ($P_0$) and to ice ($P_1$) computed from Eq. (4) and (5), and the three-Gaussian histogram fit obtained on image G2-s1.

typical computing time of the energy-based segmentation of a $1000^3$ voxel image is 5 h on a desktop
computer with a single processor (2.7 GHz).

The energy function $E$ used in the present work is composed of two components: a data fi-
delity term $E_v$ and a spatial regularisation term $E_s$. The definition of $E$ is similar to that pro-
posed by Hagenmuller et al. (2013) for the binary segmentation of impregnated snow samples
(air/ice/impregnation product), except that the data fidelity term is, here, adapted to the process-
ing of air/ice images. This term assigns penalties for classifying a voxel into ice or air, according to
its local grayscale value. Qualitatively, assigning to air a voxel with a grayscale value close to the
attenuation peak of ice "costs more" than assigning it to ice. Quantitatively, we define $E_v$ as follows:

$$E_v(L) = v \cdot \sum_i \Big( (1 - L_i) \cdot P_0(I_i) + L_i \cdot P_1(I_i) \Big) \tag{3}$$

where $L_i$ is the segmentation label (0 for air, 1 for ice) for voxel $i$, $I_i$ is its grayscale value, $P_0$ is the
proximity function to air, and $P_1$ is the proximity function to ice. This energy is scaled by the volume
$v$ of one voxel. The proximity functions quantify how close a grayscale value is to the corresponding
material. They are defined from the three-Gaussian fit (Eq. 2) adjusted on the grayscale histogram
as follows:

$$P_0(I) = \begin{cases} 1 \text{ if } I < \mu_{air} \\ \min\left(1, e\mathcal{N}(\mu_{air}, \sigma)(I)\right) \text{elsewhere} \end{cases} \tag{4}$$

$$P_1(I) = \begin{cases} 1 \text{ if } I > \mu_{ice} \\ \min\left(1, e\mathcal{N}(\mu_{ice}, \sigma)(I)\right) \text{elsewhere} \end{cases} \tag{5}$$

with $e = \exp(1)$ (Fig. 5).

The spatial regularisation term $E_s(L)$ is defined as $r \cdot S(L)$, with $S(L)$ the surface area of the
segmented object $L$ and $r$ ($r \geq 0$) a tunable parameter with the dimension of a length. Accounting
for this regularisation term in the energy leads to penalising large interface areas: a voxel with an



intermediate gray value is segmented so that the interface air/ice area is minimised. The parameter
$r$ assigns a relative weight to the surface area term in the total energy function $E$, and can be in-
terpreted as the minimum radius of protuberances preserved on the segmented object (Hagenmuller
et al., 2013). This regularisation term minimising the ice/air interface is of particular interest for ma-
terials such as snow where metamorphism naturally tends to reduce the surface and grain boundary

energy. Such processes are known to be particularly effective on snow types resulting from isother-
mal metamorphism. For other snow types, such as precipitation particles, faceted crystals or depth
hoar, the surface regularisation term is expected to perform well in recovering the facet shapes, but
may induce some rounding at facet edges.

### 2.3   Surface area computation

Flin et al. (2011) evaluated three different approaches to compute the area of the ice-pore interface
from 3D binary images: the stereological approach (e.g. Torquato, 2002), the marching cubes ap-
proach (e.g. Hildebrand et al., 1999) and the voxel projection approach (Flin et al., 2005). These
authors showed that the three approaches provide globally similar results, but each possesses its
own inherent drawbacks: the stereological approach does not handle anisotropic structures properly,

the marching cubes tends to overestimate the surface, and the voxel projection method is highly
sensitive to image resolution. In the present work, in order to estimate whether variations of SSA
due to different surface area computation approaches are significant compared to the effect of bi-
nary segmentation, we tested three different methods to quantify the surface area: the stereological
approach, the marching cubes approach and the Crofton approach. We did not evaluate the voxel

projection method (Flin et al., 2005) because its implementation is sophisticated and the required
computation of high-quality normal vectors is excessively time-consuming if used only for surface
area computation.

### 2.3.1   Model based stereological approach

Stereological methods derive higher dimensional geometrical properties, as density or SSA, from
lower dimensional data. The key idea is to count the intersections of the reference material with
points or lines. Prior to the development of X-ray tomography, so-called model based methods were
used. These models assume certain geometric properties of the object being studied, such as the
isotropy of the material (Edens and Brown, 1995). They are now replaced by design-based methods
that do not require any prior information on the studied object but require denser sampling of the
object (Baddeley and Vedel Jensen, 2005; Matzl and Schneebeli, 2010).

Here, we used two variants of the stereological method by measuring the intersection of lines
in a 3D-volume. The first method was by counting the number of interface points on linear paths
aligned with the three orthogonal directions. The surface area is then twice the number of intersec-
tions times the area of a voxel face. A surface area value is obtained for each direction. With the





microtomographic data presented in this paper, the 2D sections are virtual and do not correspond to physical surface sections of the sample. This corresponds to a model-based stereological method since isotropy of the sample is assumed, we call it in the following "stereological".

In addition, we used the mathematical formalism provided by the Cauchy-Crofton formula that explicitly relates the area of a surface to the number of intersections with any straight lines (Boykov and Kolmogorov, 2003). Instead of using only three orthogonal directions of the straight lines, we used 13 and 49 directions, and the improved approximations based on Voronoi diagrams proposed by Danek and Matula (2011). This method comes close to a design based stereological method, as the volume (and direction) is almost exhaustively sampled. We refer to this area computation method to as the Crofton approach.

### 2.3.2 Marching cubes approach

The marching cubes approach consists in extracting a polygonal mesh of an isosurface from a three-dimensional scalar field. Summing the area contributions of all polygons constituting the mesh provides the surface area of the whole image. We used a homemade version of the algorithm developed by Lorensen and Cline (1987). It computes the area of the 0.5-isosurface of the binary image without any further processing of the image. Our version of the algorithm is adapted to compute only the surface area without saving all the mesh elements that are required for 3D visualisation.

## 3 Results

In this section, the methods to compute the area of the ice-air interface are evaluated first, since this evaluation can be performed on reference objects whose area is theoretically known, without accounting for the interplay with the binary segmentation method. The Crofton approach, which is shown to perform best, is selected for the rest of the study. The sensitivity of density and SSA to the parameters of the sequential filtering and energy-based segmentations on the entire set of snow images is then investigated. Finally the variability of SSA due to numerical processing is compared to the variability of SSA due to snow spatial heterogeneity and scanning resolution.

### 3.1 Surface area estimation

An oblate spheroid (or ellipsoid of revolution) with symmetry axis along $z$ was chosen as a reference object to compare the different surface area computation methods. An anisotropy of 0.6 was considered (ratio between the dimensions of $z$- and $(x,y)$-semi-axes), and spheroids of different sizes were used to evaluate the impact of the discretization on the surface area computation. Figure 6 shows that the surface area calculated with the Crofton approach is in excellent agreement with the theoretical area: for sufficiently large spheroids, i.e. surface area larger than 200 voxels$^2$, the relative error is less than 1% for the Crofton approach with 49 different directions and 2% for the Crofton approach





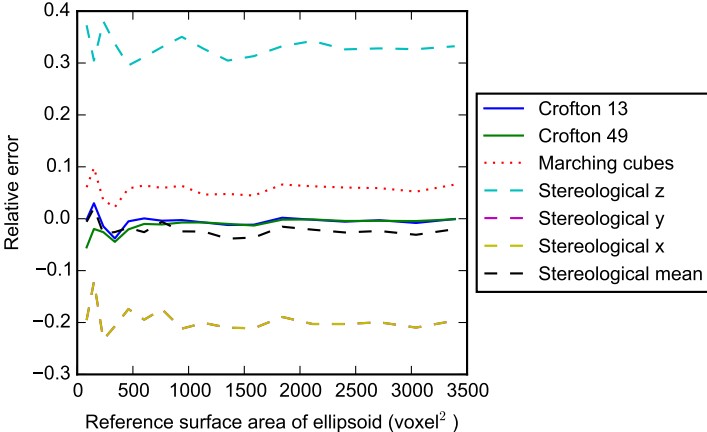

**Figure 6.** Surface area of an oblate spheroid, obtained with different calculation methods. The spheroid has a horizontal (x, y) semi-axis $a$ varying in $[3, 19]$ voxel and a vertical (z) semi-axis $c = 0.6 \cdot a$. The reference surface area of the spheroid is computed analytically. The Crofton approach is computed with 13 or 49 different directions. As in Fig. 10, 11, 12 and 13, the relative error is calculated as the computed value minus the reference value, divided by the reference value.

with 13 different directions. Adding more directions does not significantly improve the accuracy of the Crofton approach while it increases the computation time. The marching cubes approach system-

atically overestimates the surface area by about 5% due to the presence of artificial stair-steps in the triangulation of the isosurface. As expected, the stereological method shows scatter in the results obtained between the $z$- and the $(x, y)$ components. The mean value of the three components provides a fair estimation of the surface area with a relative error of about 2% compared to the theoretical value. These observations on the stereological and marching cubes approaches corroborate previous

results obtained by Flin et al. (2011) on snow images.

The different surface area computation methods were then evaluated on the entire set of snow images segmented with the energy-based method ($r = 1$). According to the results obtained on the spheroid, the Crofton approach with 13 directions was chosen as a reference. As shown in Fig. 7, the SSA obtained with the direction-averaged stereological method is in excellent agreement with

the value provided by the Crofton method. The results of the marching cubes method are in fair agreement but show a systematic over-estimation of the SSA (+6% average relative deviation).

In summary, all presented area computation method showed consistent results. The Crofton approach showed the best accuracy on an artificial anisotropic structure whose surface area is theoretically known. The stereological approach is negatively affected by strong anisotropy of the imaged

structure. However on the tested snow images, the structural anisotropy is low and this method is





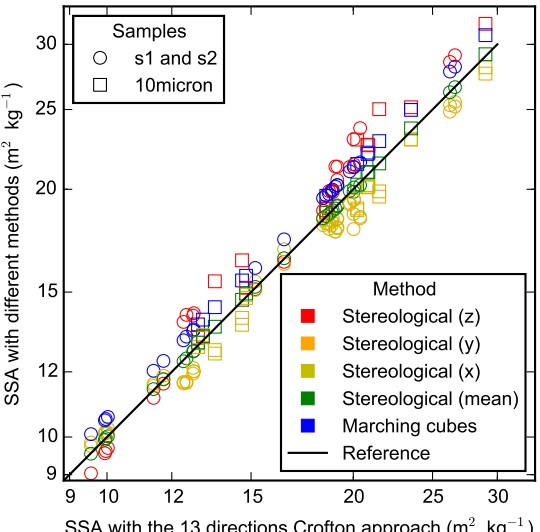

**Figure 7.** Comparison of the SSA obtained with different surface area calculation methods on all images binarized with the energy-based segmentation ($r = 1$ voxel). The root mean square difference between the SSA computed as the direction-average of the stereological method and the SSA computed with the Crofton approach is 0.008 m$^2$ kg$^{-1}$. This difference is 1.13 m$^2$ kg$^{-1}$ for the marching cubes approach. The black line represents the 1:1 line.

in excellent agreement with the Crofton approach. The simple marching cubes approach presented here (without additional filtering or pre-smoothing of the binary image) overestimates the specific surface on the order of 5%. For the following analysis of the sensitivity to binary segmentation, the SSA is computed via the Crofton approach with 13 directions.

**3.2 Sequential filtering**

The binary image resulting from the sequential filtering approach depends on: (1) the standard deviation $\sigma$ of the Gaussian filter, (2) the threshold value $T$, (3) the size $d$ of the post-processing morphological filters (opening/closing). As shown in Fig. 8, both SSA and density are sensitive to these segmentation parameters. The relation between SSA and density, on the one hand, and $\sigma$ and

$d$, on the other hand, depends significantly on the chosen threshold. Thus, in the following, we first investigate the dependence of SSA and density on the threshold, and then analyse the effects of $\sigma$ and $d$ with a threshold obtained with the mixture model of Hagenmuller et al. (2013).





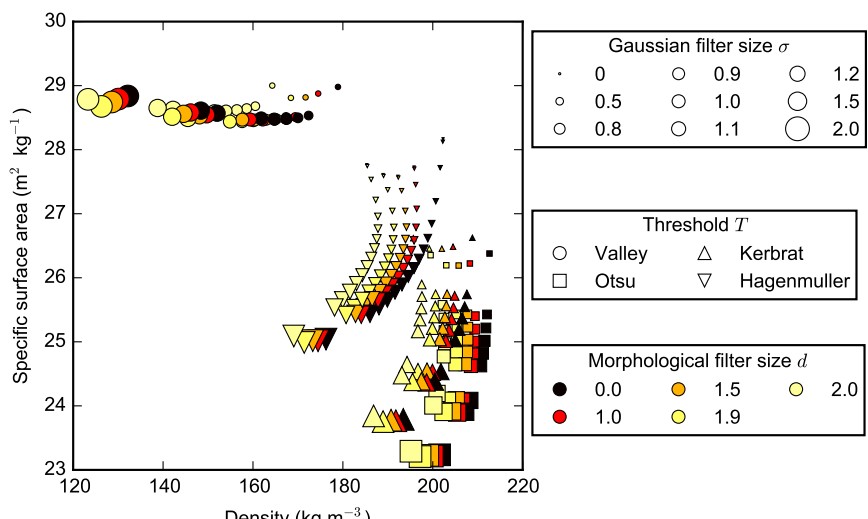

**Figure 8.** SSA and density of image G2-s1 obtained with sequential filtering for different segmentation parameters.

### 3.2.1 Choice of threshold

The threshold $T_{mask}$ obtained with the two-Gaussian fit of the grayscale histogram computed on the low-intensity gradient zones is chosen as a reference since this value is not affected by the fuzzy transition artefact. This reference threshold ranges between 38800 and 39500 for the different scanned images (Fig. 9). The mean values of the attenuation peaks of air and ice are $\overline{\mu}_{air} = 35800$ and $\overline{\mu}_{ice} = 42600$, respectively. Hence, the variations of the reference threshold value remain small compared to the contrast between the two attenuation peaks $\overline{\mu}_{ice} - \overline{\mu}_{air} = 6800$. However, these variations clearly indicate, once again, that a unique threshold value cannot be used for all images. These variations could be explained by slight variations in the X-ray source energy level due to slight temperature changes, or to deviations from the Beer-Lambert attenuation law depending on the total ice content of the sample.

As shown in Fig. 9, the computed threshold depends significantly on the determination method. These variations affect in turn the density extracted from the binary image (Fig. 10a). Note that the scatter on density due to the choice of the threshold remains the same even if a Gaussian filter is applied on the grayscale image before thresholding (Fig. 10a). The SSA values are also affected by the threshold determination method, but to a smaller extent since the threshold value tends to





affect density and total surface area in the same proportion (Fig. 10b). The variation of SSA due to
smoothing is much more important than those due to the choice of the threshold (Fig. 10b).

In detail, the valley method systematically overestimates the threshold value, leading to a systematic underestimation of the snow density by about 10 kg m$^{-3}$ on average. Otsu's method tends to underestimate the threshold value, leading to an overestimation of the snow density by about 6 kg m$^{-3}$ on average. Kerbrat's method tends to underestimate the threshold value, leading to an
overestimation of the snow density by about 4 kg m$^{-3}$ on average. Note that the density overestimation with Kerbrat's method is more pronounced on low-density snow samples scanned with a 18 $\mu$m resolution. Lastly, the method introduced by Hagenmuller et al. (2013) slightly underestimates the threshold value, and therefore overestimates the snow density with a mean absolute difference of about 2 kg m$^{-3}$ compared to the reference.

In summary, the threshold value obtained with the valley method, a method widely used in the snow community, clearly leads to an underestimation of snow density. The mixture models of Kerbrat et al. (2008) or Hagenmuller et al. (2013), which assume that noise is Gaussian distributed, provide a threshold value in good agreement with the reference method. The model of Hagenmuller et al. (2013), which explicitly accounts for the fuzzy transition between materials, yields the thresh-
old which is the closest to the reference value obtained on the masked image.

### 3.2.2   Gaussian filtering

The sensitivity of density and surface area to the standard deviation $\sigma$ of the Gaussian smoothing kernel is shown on Fig. 11. The segmentation was performed with the threshold $T_{hagen}$ derived with the method of Hagenmuller et al. (2013).

Depending on the sample, density varies in the range [-8, +2]% (compared to the value obtained without smoothing) when $\sigma$ is increased from 0 to 20 $\mu$m (Fig. 11a). Density appears to be insensitive to $\sigma$ when $\sigma$ is much lower than the voxel size. For larger values of $\sigma$, an average decrease of density with $\sigma$ is observed due to the fact that snow structure is generally convex and smoothing tends to erode convex zones. Slight increase of density with $\sigma$ is observed for $\sigma > 5$ $\mu$m for samples M1-1,
M1-3 and M2-2. These samples are the most faceted snow samples exhibiting a large proportion of flat surfaces (Fig. 1), which explains the different variation of density with $\sigma$. Systematic differences can also be noted between the images with a resolution of 10 $\mu$m and 18 $\mu$m. At a resolution of 10 $\mu$m, a fast decrease of density is observed for $\sigma$ in the range [3, 6] $\mu$m. This regime is absent at a resolution of 18 $\mu$m. For larger values of $\sigma$, the evolution of density is then similar for the two
resolutions, and depends on the snow type. This difference is attributable to a stronger noise in the 10 $\mu$m images, which results in local grayscale variations that are generally smoothed out when $\sigma > 6$ $\mu$m.

The computed surface area significantly decreases when $\sigma$ increases (Fig. 11b). Relative variations up to 50% are observed. On the 10 $\mu$m images and with $\sigma$ in the range [3, 6] $\mu$m, the surface





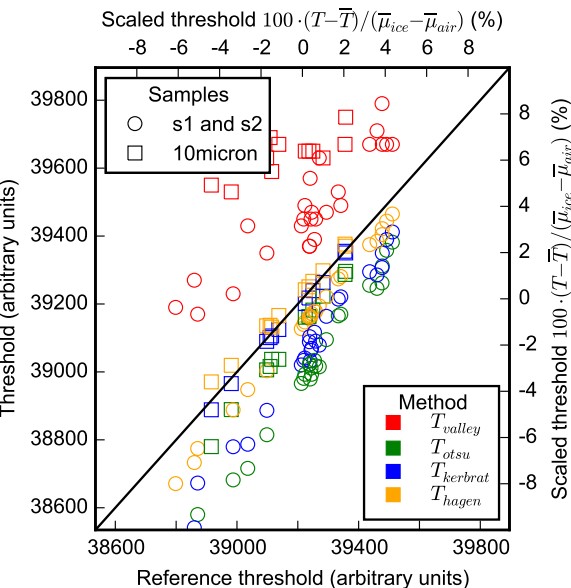

**Figure 9.** Sequential filtering segmentation: threshold values obtained on the entire set of images with the different intensity models. The black line represents the 1:1 line. The obtained thresholds $T$ are also expressed as a function of the mean threshold $\overline{T}$ and the mean contrast $\overline{\mu}_{ice} - \overline{\mu}_{air}$ between air and ice, obtained with the reference method.

area decreases rapidly when $\sigma$ increases. These variations probably correspond to the progressive smoothing of noise-induced fluctuations on the interface. For larger values of $\sigma$, the surface area decreases much more slowly, which corresponds to the progressive smoothing of real microstructural details. On the 18 $\mu$m images, these two regimes cannot be distinguished because the overall surface area is less affected by noise artefacts and only the smoothing of real structural details is observed.

The same variations with $\sigma$ can be observed on SSA since the variations of density with $\sigma$ are small compared to the variations of the surface area. Note that the absolute values of density and SSA for the different scanned snow images are indicated on Fig. 15.

### 3.2.3   Morphological opening/closing

Figure 12 shows the relative variation of density and surface area obtained with morphological filters
of different sizes $d$. Note that the values of $d$ are constrained by the voxel grid and are thus discrete ($1$, $\sqrt{2}$, $\sqrt{3}$, $2$ voxel, etc.). The opening and closing filters delete holes in the ice matrix or ice elements in the air, of a typical size $d$. Therefore, the surface area decreases when $d$ increases. Density, on the contrary, is not very sensitive to $d$. When no Gaussian filter is applied to the grayscale





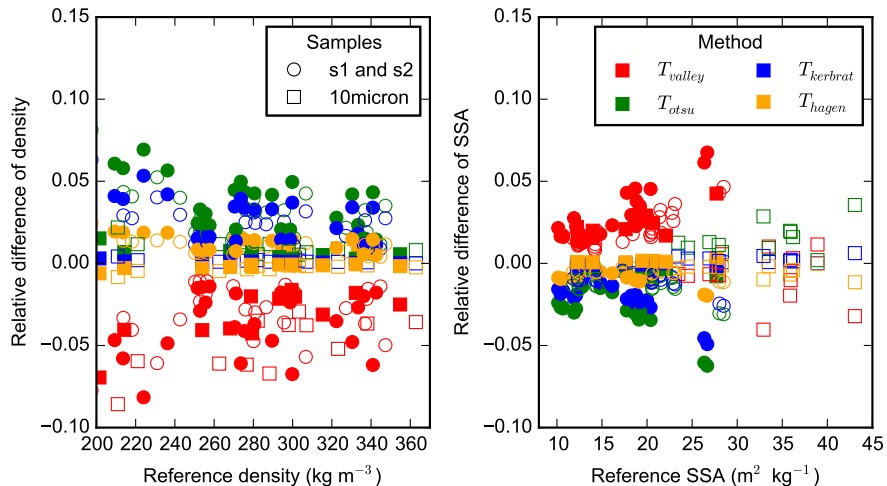

**Figure 10.** Sequential filtering segmentation: relative variation of density (a) and SSA (b) computed with different threshold-determination methods, with respect to the reference values computed with $T_{mask}$. Void (respectively solid) markers correspond to values obtained without any smoothing (respectively with a Gaussian filter of standard deviation $\sigma = 1$ voxel). The legends apply to both subplots.

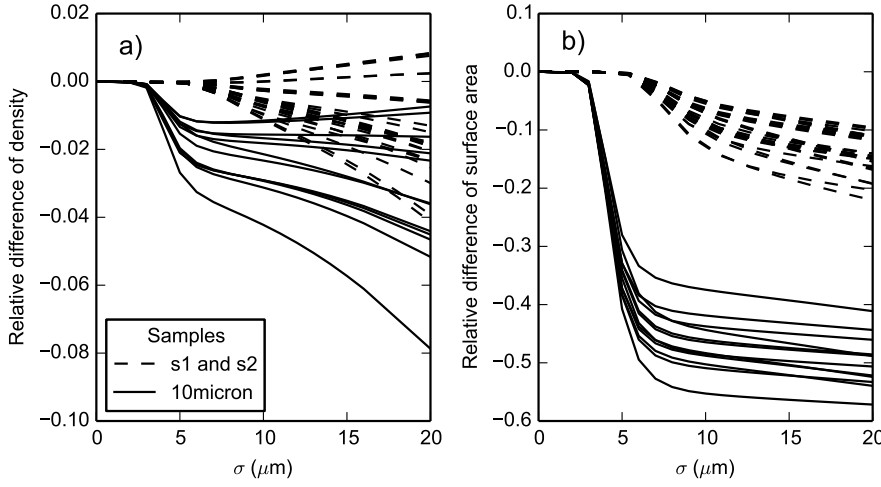

**Figure 11.** Sequential filtering segmentation: relative variations of density (a) and surface area (b) with the size $\sigma$ of the Gaussian filter. The variations are calculated with respect to density and surface area obtained without smoothing filter ($\sigma = 0$).





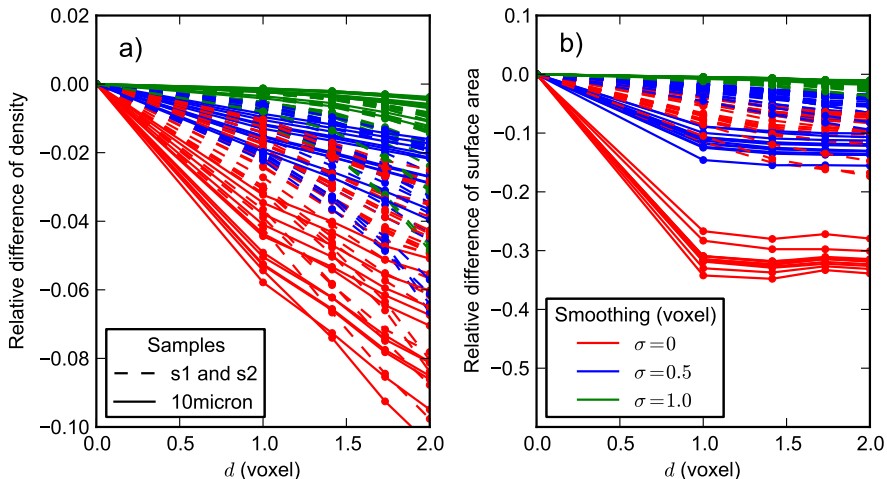

**Figure 12.** Sequential filtering segmentation: relative variations of density (a) and surface area (b) as a function of the morphological filter size $d$ for different values of the Gaussian filter size $\sigma$. The legends apply to both subplots.

image, thresholding yields a lot of small details in the binary image, which enhances the effect of
morphological filters (Fig. 12b). When the image is already smoothed by Gaussian filtering, the morphological filters have less effect on the overall density and specific surface area. However, note that using a Gaussian filter of standard deviation $\sigma$ does not guarantee the complete absence of details "smaller" than $\sigma$. Certain algorithms based on the binary images, such as grain segmentation (e.g. Theile and Schneebeli, 2011; Hagenmuller et al., 2014) are highly sensitive to the presence of
residual artefacts in the ice matrix and require the use of these additional morphological filters.

### 3.3 Energy-based approach

The binary image image resulting from the energy-based approach depends on the parameter $r$ which controls the smoothness of the segmented object. The other parameters involved in the volumetric term $E_v$ of the segmentation energy are directly derived from the three Gaussian mixture model (see
Sect. 2.2.1).

As shown in Fig. 13a, the density of the segmented object slightly varies with $r$. On the 10 $\mu$m images, the evolution of density with $r$ is not monotonic but relative variations remain limited in the range [-4, +1]%. On the 18 $\mu$m images, density clearly decreases when $\sigma$ increases. This higher sensitivity of density to $r$ on the 18 $\mu$m images can be explained by the fact that the fuzzy transition
appears to be larger than on the 10 $\mu$m images, which leads to a higher indetermination of the exact position of the interface between ice and air in this moderate intensity gradient zone (Fig. 14).




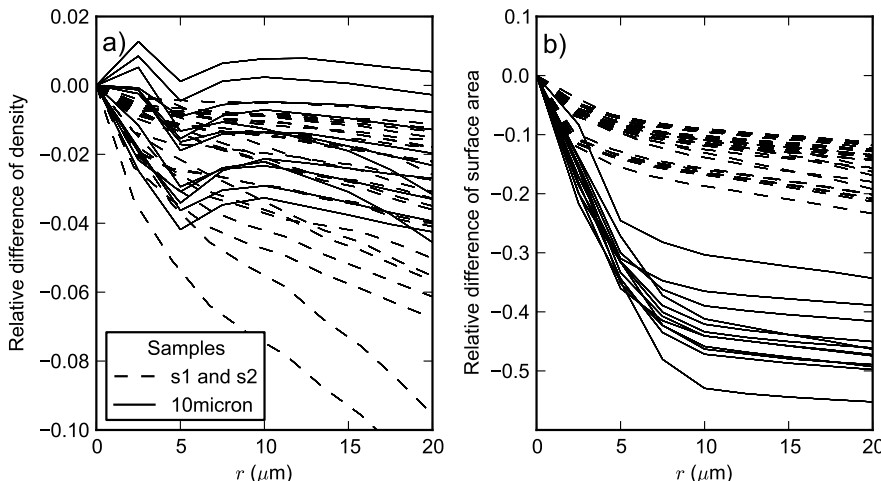

**Figure 13.** Energy-based segmentation: relative variations of density (a) and surface area (b) as a function of parameter $r$.

As shown in Fig. 13b, surface area is more sensitive to $r$ than density, and decreases significantly when $r$ increases. Two regimes can be distinguished. For low values of $r$ in the range $[0, 10]$ $\mu$m, the surface area decreases rapidly when $r$ increases. For larger values of $r$ in the range $[10, 20]$ $\mu$m,

the decrease of the surface area with $r$ is much slower, and displays an almost constant slope. As discussed by Hagenmuller et al. (2013), this second regime is due to real details of the snow structure being progressively smoothed out, and is indicative of a continuum of sizes in structural details of snow microstructure. The distinction between the two regimes is more pronounced on the 10 $\mu$m images which are more affected by noise (Fig. 14).

**3.4 Comparison between images and methods**

The sensitivity of surface area to the parameters $\sigma$ and $r$ is similar, but the energy-based and sequential filtering approaches are conceptually different (Fig. 11 and 13). The Gaussian filter smoothes high-frequency intensity variations with a small amplitude, independently of the subsequent binary thresholding. Small details due to noise artefacts remaining in the binary image are then deleted

independently of the initial grayscale value by applying morphological filters. The energy-based approach smoothes the segmented object so that the ice-air interface area is minimised while respecting at best the grayscale intensity model. Hence, the grayscale smoothing and morphological filtering are somehow done simultaneously with thresholding in the energy-based approach. In addition, the Gaussian filter is "grid-limited": as shown in Fig. 11b, this filter does not affect the segmented object

if $\sigma$ is to small compared to the voxel size. In contrast, in the energy-based approach, smoothing of





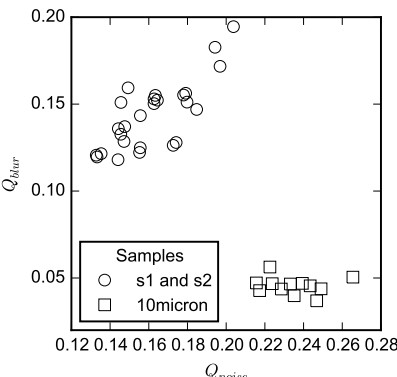

**Figure 14.** Relative importance of the artefacts due to noise ($Q_{noise}$) and to the fuzzy transition ($Q_{blur}$). $Q_{noise}$ corresponds to the ratio between the standard deviation of the Gaussian distributions fitted on the attenuation peaks and the difference between the peak attenuation intensity of ice and air (see Sect. 2.2.1). $Q_{blur}$ corresponds to the area of the Gaussian representing the fuzzy transition in Hagenmuller's mixture model (see Sect. 2.2.1).

the ice-air interface occurs even for very low values of $r$ (Fig. 13b) because voxels with a grayscale value close to the threshold between ice and air can be segmented as air or ice without much change in the data fidelity term $E_v$ but with a clear change in the surface term $E_s$. The parameter $r$ defines the largest equivalent spherical radius of details preserved in the segmented image, whereas $\sigma$ does

not directly correspond to the size of the smallest detail.

Figure 15 shows density and specific surface area computed on the entire set of snow images segmented with the sequential filtering approach ($\sigma = 1.0$ voxel, $T = T_{hagen}$, $d = 1.0$ voxel) and the energy-based approach ($r = 1.0$ voxel). The "smoothing" parameters ($\sigma$ and $r$) were chosen equal to 1.0 voxel since this value roughly corresponds to the transition beyond which the computed surface

area starts to vary slowly with $\sigma$ and $r$ (Fig. 11b, 13b), and therefore provides the segmentations that best preserve the smallest snow details while deleting most of noise-induced protuberances. As already pointed out, this transition is clear on the 10 $\mu$m images, but less evident on the 18 $\mu$m images. To be consistent, however, and to ensure that all noise artefacts are smoothed out, values of $r, \sigma = 1$ voxel were used in all cases.

It is observed that the two approaches generally produce similar results in terms of density (root mean square deviation between the two segmentation methods is 6 kg m$^{-3}$) and specific surface area (root mean square deviation of 0.7 m$^2$ kg$^{-1}$). The largest differences are observed for the snow types presenting the highest SSA. In general, the density provided by the sequential filtering is slightly larger than that computed with the energy-based approach. The opposite difference is observed for

SSA.



Scatter can be observed even between the density and SSA derived from images coming from the same snow block, due probably to the existence of spatial heterogeneities with the blocks and and the difference of image quality. The averages of standard deviations calculated for each snow block are 10.7 kg m$^{-3}$ and 1.1 m$^2$ kg$^{-1}$ for density and SSA, respectively (calculated with the energy-based approach). This intra-block variability nevertheless appears to be limited compared to the inter-block variability (46.5 kg m$^{-3}$ for density and 4.7 m$^2$ kg$^{-1}$, see Fig. 15), and is on the same order as the variability due to the image processing technique (see above).

Lastly, systematically larger density and SSA values are found on the images scanned with a 10 $\mu$m resolution, compared to the images with a 18 $\mu$m resolution. It could be argued that this difference is due to a better imaging of small details with a lower voxel size. However, as already noticed, the 10 $\mu$m images also present stronger noise artefacts (Fig. 14), and it is difficult to assess whether the effective resolution of these images is, in practice, finer than the one of the 18 $\mu$m images. Note that the root mean square difference between the density (respectively SSA) computed on the images s1 and s2 is 3.8 kg m$^{-3}$ (respectively 0.15 m$^2$ kg$^{-1}$), which is much lower than the intra-block variability (including the 10 $\mu$m images). This observation indicates that the hardware setup of the tomograph and the subsequent image quality or resolution can significantly affect the measured density and SSA.

## 4  Conclusions and discussions

We investigated the effect of numerical processing of microtomographic images on density and specific surface area derived from these data. To this end, a set of 38 X-ray attenuation images of non-impregnated snow were analysed with different numerical methods to segment the grayscale images and to compute the surface area on the resulting binary images.

The segmentation step is not straightforward because the grayscale images present noise and blur. It is shown that noise artefacts can significantly affect the computed SSA, and that the fuzzy transition between ice and air can have a strong impact on the computed density.

The sequential filtering approach critically depends on the threshold used to separate ice and air. The grayscale histogram on low-intensity gradient zones presents two disjoint attenuation peaks, whose characteristics are not affected by blur. The threshold derived from this method was used as a reference to evaluate other methods based on the analysis of the grayscale histogram of the entire image. The mixture models which consist in decomposing the histogram into a sum of Gaussian distributions are shown to be accurate. On the contrary, the local minimum method is shown to be unsuitable in general.

Smoothing induced by the Gaussian and morphological filters in the sequential approach, or by accounting for the surface area term in the energy-based method, efficiently remove noise artefacts from the segmented binary image. Morphological filters applied on the binary image in the sequential





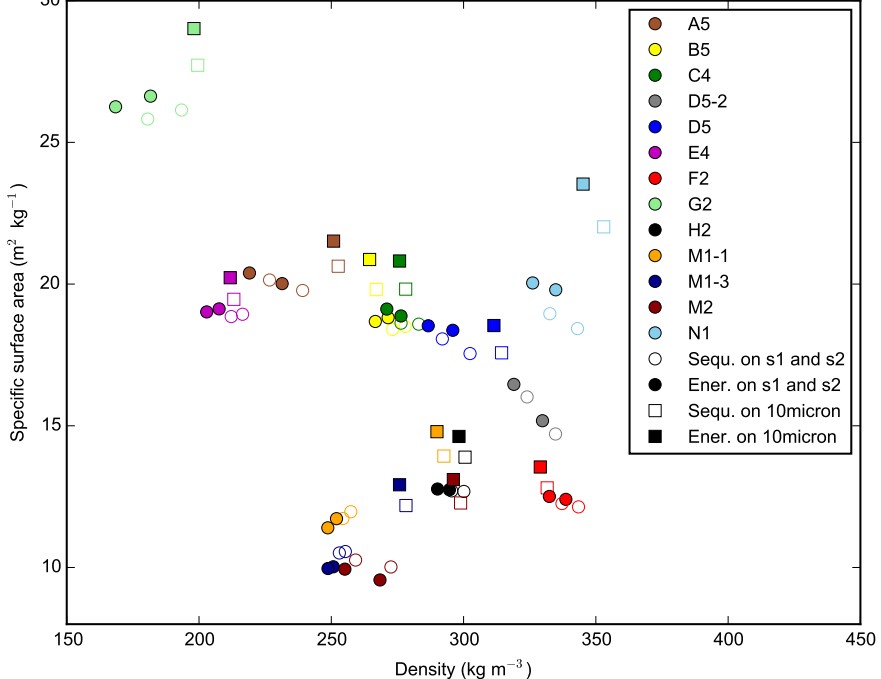

**Figure 15.** Specific surface area as a function of density for the entire set of images and the two different binary segmentation methods. The sequential filtering ("Sequ." in the legend) was applied with $\sigma = 1.0$ voxel, $T = T_{hagen}$ and $d = 1.0$ voxel. The energy-based approach ("Ener." in the legend) was applied with $r = 1.0$ voxel. The surface area was computed with the Crofton approach.

approach miss the initial gray value information. However, it seems that their effect is negligible if the applied Gaussian filter is strong enough. The smoothing can also induce the disappearance of real structural details contributing to the overall SSA. The transition between smoothing of noise and smoothing of real details can be well estimated on the curve showing the evolution of SSA as a function of $\sigma$ or $r$. However, due to the influence of noise, it remains difficult to assess the potential contribution to the SSA of structural details of size smaller than the voxel size. It has previously been shown that the SSA measured with gas adsorption technique, which has a molecular resolution, is in good agreement with the SSA measured with microtomography for aged natural snow (Kerbrat et al., 2008). This observation corroborates the idea that the surface of aged snow is smooth up to a scale of about tens of microns, and that if smaller structures are present they do not contribute significantly to the overall SSA (Kerbrat et al., 2008). To further investigate this issue on recent



snow and to disregard any additional influence of the measuring technique except the resolution, the use of new tomographic systems with very high resolutions of about 1 $\mu$m would be necessary.

The formalism of the energy-based segmentation could enable to add more advanced criteria in the segmentation process, such as the maximisation of the grayscale gradient at the segmented interface (Boykov and Jolly, 2001), the minimisation of the curvature of the segmented object (El-Zehiry and Grady, 2010), or the spatial continuity in time-series of 3D images (Wolz et al., 2010). In this study, only criteria on the local grayscale value and on the surface area of the segmented object were used. The advantage of this method is that the parameter $r$ formally defines an effective resolution of the segmented image. In contrast, the standard deviation $\sigma$ of the Gaussian smoothing kernel in the sequential approach does not explicitly define the smallest structural detail in the segmented image. In practice, however, both methods provide very similar results on the tested images in terms of density and SSA, provided appropriate parameters are chosen.

Comparison between the presented area computation methods showed similar results when applied to a synthetic image or to the set of snow images. On the synthetic image (oblate spheroid), the Crofton approach computes the surface area with highest accuracy (less than 2% for sufficiently large spheroids) whereas the stereological approach is negatively affected by strong anisotropy of the imaged structure and the unfiltered marching cubes approach overestimates the specific surface on the order of 5%. Stereological methods using more complex test lines, such as a cycloids, can compensate for the effect of anisotropy if the snow sample exhibits isotropy in a certain plane, which is often the case for the stratified snowpack (Matzl and Schneebeli, 2010). However on the tested snow images, the surface anisotropy is low and the stereological method is in excellent agreement with the Crofton approach. The unfiltered marching cubes approach still overestimates the specific surface on the order of 5%. Note that methods have been developed to overcome this overestimation problem of the marching cubes approach, such as the use of gray levels or smoothing (Flin et al., 2005). However, these methods may create other artefacts depending on the image considered, such as systematic underestimation of the surface (Flin et al., 2005), and were not evaluated here.

The comparison of the sequential filtering and energy-based methods shows that density and SSA can be estimated from X-ray tomography images with a "numerical" variability of the same order as the variability due to spatial heterogeneities within one snow layer and to different hardware setups.

## 5 Recommendations

A few recommendations to derive density and SSA from micro-tomographic data are summarized below:

- *Surface area computation.* The unfiltered marching cubes approach systematically overestimates the surface area and should thus be avoided. Counting intersections with test lines of



different orientations (at least in the three axes $x$, $y$ and $z$) provides an efficient way to compute the surface area and properly accounts for structural anisotropy.

– *Threshold determination.* The value of the threshold depends on the tomograph configuration but also potentially on the scanned sample. A constant value for a time-series does not necessarily prevent from density deviations due to beam hardening. Visual inspection of the histogram or the "valley method" do not always provide consistent threshold values. The fit of Gaussian distributions on the histogram provides an automatic and satisfactory method to determine an appropriate threshold value. However, all methods need visual inspection and comparison with the grayscale image.

– *Smoothing of grayscale image.* Smoothing of the grayscale image, such as the convolution with a Gaussian kernel, is required to reduce noise artefacts but also reduces the effective resolution of the image by deleting structural details that could contribute to the overall SSA. The filter, and in particular the standard deviation $\sigma$ of the applied Gaussian kernel, expressed in $\mu$m, should be systematically mentioned if SSA values derived from tomographic data are presented. Indeed, SSA is a decreasing function of the effective resolution even if the resolution is larger than the nominal voxel size. This function is expected to become constant only for sufficiently small resolutions depending on the snow type.





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
