# Peer review of "Sensitivity of snow density and specific surface area measured by microtomography to different image processing algorithms"

_The Cryosphere, 2015_

## Referee Comment (RC1) · Anonymous Referee #1 · 15 Mar 2016

This is a very good designed study. The authors compared different image processing algorithms to calculate density and specific surface area (SSA) using microtomography image of snow samples, then they summarized the influence of each procedure on the calculation results. In addition, they suggested the several recommendations to derive density and SSA from microtomographic data. I do not doubt that their study will give large contributions to advance on the study in snow microstructure using $\mu$-CT. Therefore, it is worthy of publication in The TC. The logic in the manuscript is well organized and I did not find any contradiction. Before acceptance, only one thing, which is shown in the comment, should be confirmed.

Comments Figure 4: The value of L1 of (a) (0.006) is smallest, is it true? From the

figures, I guess the value of L1 of (a) should be largest.

---

## Referee Comment (RC2) · Anonymous Referee #2 · 29 Mar 2016

The content of the paper is suitable for publication in The Cryosphere. I believe the approach is generally sound and conclusions relevant. However, there are a few areas that would benefit from additional clarification. In particular, the surface energy concept applied to the CT analysis, in my opinion, needs some additional explanation. I have attached an annotated pdf copy of the paper with my comments and suggestions.

Please also note the supplement to this comment:
http://www.the-cryosphere-discuss.net/tc-2015-217/tc-2015-217-RC2-supplement.pdf

[revised manuscript text omitted]

**Notes**

**1-1** This sort of statement that directs a reader to the main article text in order to gain the information is generally discouraged in an abstract. Ideally, an abstract should be essentially self contained.

**1-3** Alternate definition surface/volume

**2-1** I don't the reason that this might be the case. Is a short explanation possible?

**2-4** Some of these, tomography for instance, which is the thrust of this paper, would use the alternate units.

**3-1** Why is this relevant to binary segmentation ? Changes taking place during the time of the scan?

**3-4** Due to curvature? How does the the tendency toward minimization influence the current configuration? There would be no metamorphism for the impregnated samples.

**3-5** I think this needs some explanation. Snow is not necessarily at a minimum. It is tending toward it in the absence of an imposed temperature gradient.

**5-2** Same for both?

**7-1** Fig. 2 as well?

**8-1** mu is not defined. (Attenuation peaks)

**8-4** Attenuation corresponds to occurrence ratio? Not clearly defined.

**8-8** What do these terms represent? They are not defined.

**11-1** I do not see how this influences the "instantaneous" scan. This needs to be more fully discussed.

**13-1** I do not see the stereological y. Is it equivalent to the x and superimposed?

**13-2** voxel?

**15-1** I find this a somewhat confusing figure to interpret with the legends as presented.

**15-2** Is density not simply a surrogate for the solid volume, assuming ice as the constituent? Isn't it really the volume that you're measuring, independent of material?

**20-1** Earlier r was given in terms of voxel?

**21-1** I do not see where lamda is defined.

---

## Author Comment (AC1) · 11 Apr 2016

We thank the referees for their positive feedback on this work. The responses to their comments can be found in the supplement pdf file.

The authors.

Please also note the supplement to this comment:
http://www.the-cryosphere-discuss.net/tc-2015-217/tc-2015-217-AC1-supplement.pdf

---

## Author Response (AR1)

**Response to the referees (tc-2015-217)**

**Title:** Sensitivity of snow density and specific surface area measured by microtomography to different image processing algorithms.

The response to the referees is organized as follows:
- Initial comments: black
- Answers: blue
- Modification in text: red

The modifications in the revised paper are shown in red.

**Anonymous Referee #1**

This is a very good designed study. The authors compared different image processing algorithms to calculate density and specific surface area (SSA) using microtomography image of snow samples, then they summarized the influence of each procedure on the calculation results. In addition, they suggested the several recommendations to derive density and SSA from microtomographic data. I do not doubt that their study will give large contributions to advance on the study in snow microstructure using μ-CT. Therefore, it is worthy of publication in The TC. The logic in the manuscript is well organized and I did not find any contradiction. Before acceptance, only one thing, which is shown in the comment, should be confirmed.

We thank reviewer 1 for his positive and encouraging feedback.

**Comments Figure 4:** The value of L1 of (a) (0.006) is smallest, is it true? From the figures, I guess the value of L1 of (a) should be largest.

We checked the values of the L1 error presented in Figure 4. They are correct. Note that the L1 error on subfigure (a) is the error between the masked histogram and the reconstructed histogram (and NOT the initial histogram). In the other subfigures (b) and (c), the L1 errors are computed between the initial and reconstructed histograms. Note that the area below the initial histogram is scaled to 1. The very low value of L1 error in subfigure (a) indicates that, outside of the fuzzy transition between ice and air, the distribution of the grayscale value in one material is perfectly Gaussian distributed.

To avoid confusion, this is detailed in the text now.

**Anonymous Referee #2**

The content of the paper is suitable for publication in The Cryosphere. I believe the approach is generally sound and conclusions relevant. However, there are a few areas that would benefit from additional clarification. In particular, the surface energy concept applied to the CT analysis, in my opinion, needs some additional explanation. I have attached an annotated pdf copy of the paper with my comments and suggestions.

We thank reviewer 2 for his positive feedback.

1) This sort of statement that directs a reader to the main article text in order to gain the information is generally discouraged in an abstract. Ideally, an abstract should be essentially self contained.
Sentence deleted.

2) Alternate definition surface/volume.
We agree both definitions co-exist.
The alternate definition and its relation with the definition of SSA chosen here, is now detailed.

3) I don't the reason that this might be the case. Is a short explanation possible?
It is our mistake. There is no reason why the size of a monocrystalline grain should be equal to r_eq. It could be the case for a spherical grain.
Changed to "The crystal size as stereologically measured by \citet{Riche2012a} is another potential definition for grain size, a priori, independent of the other definitions mentioned above."

4) Some of these, tomography for instance, which is the thrust of this paper, would use the alternate units.
Given the accuracy of the measuring technique and the variation of rho_ice in the "usual" temperature range (at -20°C, rho_ice=919 kg/m3; at 0°C, rho_ice=916kg/m3), the relation between S/V and S/M is straightforward. There is no point to discuss in details the choice of the SSA unit.

5) Why is this relevant to binary segmentation ? Changes taking place during the time of the scan?
There is no change expected to occur during the scan (2h scanning, on already evolved snow, at low temperature -15°C).
Isothermal metamorphism tends to reduce the surface energy of snow by vapor transport. This can occur during snowfall or when the snow is on the ground. Therefore the likelihood to have a small structural detail at the snow surface with a high specific surface area is very low. This information can be used to improve the binary segmentation.

The term "minimal" is misleading. The term "limited" i.e. below a certain threshold, is more appropriate. For instance, Kerbrat et al. (2008) concluded that sublimation (Kelvin effect) and surface redistribution (diffusion in a quasi-liquid layer) are capable of smoothing the surface on a length scale of about $30\mu$m within a few hours.
Changed "the surface energy of snow tends to be minimal" to "the local surface energy of snow tends to be limited due to inherent snow metamorphism that occurred before sampling"

6) Due to curvature? How does the tendency toward minimization influence the current configuration? There would be no metamorphism for the impregnated samples.
See answer to comment 5).

7) I think this needs some explanation. Snow is not necessarily at a minimum. It is tending toward it in the absence of an imposed temperature gradient.
See answer to comment 5).

8) Same for both?
Yes, it is surprising. We expect a larger REV for density than for SSA. This is not the case in the REV sizes given in the cited papers, certainly due to different thresholds on acceptable variability (10% in Coleou et al, 2001, no explicit value given in Flin et al, 2011 but estimated to 2% from the figures in the paper) and to different snow types tested.
Changed to "on the order of 2.5$^3$~mm$^3$"

9) Fig. 2 as well?
Corrected.

10) mu is not defined. (Attenuation peaks)
Corrected.

11) Attenuation corresponds to occurrence ratio? Not clearly defined.
The grayscale distribution represents the number of occurrences of a grayscale value (in a certain range) (grayscale value = intensity = proportional to attenuation coefficient) in a certain range as a function of the grayscale value. This is clearly shown in Fig. 3. If there is a possible confusion between attenuation, intensity and grayscale value, this is now corrected.
"The grayscale value quantifies the X-ray attenuation coefficient." -> "The grayscale value or intensity $I$ quantifies the X-ray attenuation coefficient."
"convoluting the intensity field $I$" -> "convoluting the intensity field $I$ (i.e., the grayscale value)"

12) What do these terms represent? They are not defined.
Corrected.

13) I do not see how this influences the "instantaneous" scan. This needs to be more fully discussed.
See comment 5). This occurred previously to the scan.

14) I do not see the stereological y. Is it equivalent to the x and superimposed?
Yes due to symmetry, it is superimposed with x.
Annotated in the figure legend.

15) voxel?
Note (comment 18) that the size of r expressed in meters is thus different between the images with 10 or 18 microns resolution. This difference of r expressed in meters does not change the conclusion on the surface area estimators.
Corrected.

16) I find this a somewhat confusing figure to interpret with the legends as presented.
The figure contains a lot of data plotted in a concise manner. The couples (density, SSA) are plotted for various sets of three parameters (the smoothing filter, the threshold determination technique and the morphological filter). To this end, we played on size, shape and color of the plotted points to distinguish the parameters. Please be more precise if you think of a better representation.

17) Is density not simply a surrogate for the solid volume, assuming ice as the constituent? Isn't it really the volume that you're measuring, independent of material?
Yes, snow density (rho) of ice is simply rho = rho_ice * V_ice / V_tot with rho_ice density of ice, V_ice the volume of ice and V_tot the total volume of the sample. Indeed, the quantity mesured directly from the image is the volume and not density. This point comes back to the discussion on the unit of SSA. Given that this paper is intended to be published in The Cryosphere, in a special issue dedicated to the comparison of measuring techniques applied to snow, we think that the best choice is to present the results in the units most commonly used in this domain.

18) Earlier r was given in terms of voxel? Yes. The size of the voxel is the resolution so the relation between r expressed in voxel or in m is straightforward. The images s1 and s2 do not have the same resolution. For clarity purpose, we prefer to compare the effect of r on these two image types by expressing r in terms of microns instead of voxels of different sizes.

19) I do not see where lamda is defined.
See comments 12.